# The Microbiome and Metabolomic Profile of the Transplanted Intestine with Long-Term Function

**DOI:** 10.3390/biomedicines10092079

**Published:** 2022-08-25

**Authors:** Raffaelle Girlanda, Jedson R. Liggett, Meth Jayatilake, Alexander Kroemer, Juan Francisco Guerra, Jason Solomon Hawksworth, Pejman Radkani, Cal S. Matsumoto, Michael Zasloff, Thomas M. Fishbein

**Affiliations:** 1MedStar Georgetown Transplant Institute, MedStar Georgetown University Hospital, Center for Translational Transplant Medicine, Georgetown University, Washington, DC 20007, USA; 2Department of Surgery, Naval Medical Center Portsmouth, Portsmouth, VA 23704, USA; 3Department of Oncology, Lombardi Comprehensive Cancer Center, Georgetown University Medical Center, Washington, DC 20007, USA; 4Department of Surgery, Walter Reed National Military Medical Center, Bethesda, MD 20812, USA

**Keywords:** intestinal transplant, microbiome, mRNA, metabolomic, Enterococcus

## Abstract

We analyzed the fecal microbiome by deep sequencing of the 16S ribosomal genes and the metabolomic profiles of 43 intestinal transplant recipients to identify biomarkers of graft function. Stool samples were collected from 23 patients with stable graft function five years or longer after transplant, 15 stable recipients one-year post-transplant and four recipients with refractory rejection and graft loss within one-year post-transplant. *Lactobacillus* and *Streptococcus* species were predominant in patients with stable graft function both in the short and long term, with a microbiome profile consistent with the general population. Conversely, Enterococcus species were predominant in patients with refractory rejection as compared to the general population, indicating profound dysbiosis in the context of graft dysfunction. Metabolomic analysis demonstrated significant differences between the three groups, with several metabolites in rejecting recipients clustering as a distinct set. Our study suggests that the bacterial microbiome profile of stable intestinal transplants is similar to the general population, supporting further application of this non-invasive approach to identify biomarkers of intestinal graft function.

## 1. Introduction

Intestinal failure is a life-threatening condition resulting from trauma, inflammatory bowel disease, motility disorders and associated conditions that require total parenteral nutrition (TPN) for patient survival. However, prolonged TPN is associated with life-threatening complications, including liver failure, catheter-related infections and venous thrombosis with eventual loss of access [1]. In such patients, intestinal transplantation offers a valuable treatment option to restore intestinal function and resume enteral nutrition. Unfortunately, intestinal transplantation accounts for less than 0.5% of all organ transplants to date [2], and the rate of intestinal transplantation has been steadily decreasing since 2005 [3] due to the complexity of this transplant with higher incidence of rejection (up to 50%) compared to other transplants. In addition, due to the high doses of immunosuppressants required to prevent rejection, these patients are exposed to severe post-transplant infections and other immunosuppression-related complications. Key factors for long-term graft function and patient survival are early detection and treatment of rejection and the prevention of life-threatening infections.

Currently, intestinal transplant recipients are monitored with frequent clinical examinations, routine comprehensive laboratory panels to include immunosuppression levels and periodic endoscopy with biopsy of the transplanted intestine. This monitoring protocol is costly and invasive. Furthermore, occasionally the endoscopic and histological findings of patients with allograft dysfunction are inconclusive since the manifestations of rejection and those secondary to other causes, such as viral enteritis, often overlap. Therefore, novel non-invasive biomarkers are needed to improve the accuracy of post-transplant surveillance.

In recent years, the application of non-culture-based techniques, including next-generation sequencing, has expanded our knowledge of the intestinal flora and their function in the maintenance of human health. This represented an advancement from historical culture-based techniques [4]. Since the human microbiome study [5], it has become apparent that the microbial communities that colonize the intestinal tract are in a symbiotic relationship with the host and influence both homeostasis and susceptibility to disease. As a result, a microbiome-based approach has been adopted to investigate multiple disease processes such as obesity, asthma, diabetes, and inflammatory bowel disease [6] and has also been previously applied to the study of intestinal transplantation. We previously demonstrated that a shift in the microbiome from aerobic-anaerobic condition to a predominantly anaerobic condition occurs following the restoration of intestinal continuity of intestinal transplant recipients [7]. By metabolomic analysis, we also demonstrated that several pro-inflammatory features are present in the stool or ileostomy effluent of intestinal transplant patients at the time of acute rejection [8].

In the present study, we hypothesized that distinct microbial and metabolomic profiles exist in intestinal transplant patients based on the functional status of the graft and the time interval from transplant. We evaluated stool samples of intestinal transplant recipients from our center and utilized a dual microbiome and metabolomic analysis approach to characterize the gut microbiome and metabolome of stable patients with preserved long-term function (5 years or longer after transplant), stable patients with short term graft function (1 year) and unstable patients with early rejection and graft loss within one year from transplant.

## 2. Methods

### 2.1. Patient Selection

We retrospectively analyzed stool samples collected prospectively from 43 patients who underwent intestinal transplantation at our institution. This study was approved by the Institutional Review Board at Georgetown University (protocol #2004-008), and informed consent was obtained from all patients. Patients were selected according to the condition of the graft function at the time of sampling (stable versus unstable) and according to timing post-transplant (long term versus short term). Patients were thereby stratified into one of three groups:Long-term graft acceptance (Group A *n* = 23). Patients were at least five years beyond transplantation with normal intestinal allograft function, minimal immunosuppression requirement and no rejection episodes beyond the first 6 months after transplantation. These patients demonstrated normal histology on surveillance endoscopic biopsy.Short term graft acceptance (Group B *n* = 16). Patients within the first year after transplant with normal intestinal allograft function, restoration of intestinal continuity and no rejection episodes beyond the first six months. These patients demonstrated normal histology on surveillance endoscopic biopsy.Short term graft non-acceptance (Group C *n* = 4). Patients within the first year after transplantation demonstrated graft dysfunction resulting from an early rejection episode. In these patients, ongoing inflammatory changes and/or rejection were present on surveillance endoscopic biopsy of the allograft and were associated with the need for increased immunosuppression and resumption of parenteral nutritional or fluid supplementation.

Patient demographics, causes of intestinal failure and the type of grafts are included in Table 1.

### 2.2. Transplant Protocol

Intestinal transplantations and post-operative surveillance and management were conducted per institutional protocol, as previously described [9,10,11]. Patients received one of four types of intestinal graft, as dictated by presenting diagnosis and gastrointestinal anatomy pre-transplant. These grafts included isolated small intestine, a combined liver-intestine allograft, a multivisceral (MV) transplant (liver-intestine *en-bloc* with the foregut) or a modified multivisceral (MMV) transplant, which is an MV without liver [1,12]. The colon was included in the graft and transplanted in continuity with the small intestine in patients with totally absent or dysfunctional colon pre-transplant [13]. Patients underwent routine loop ileostomy creation at the time of transplant for post-transplant monitoring, with planned restoration of intestinal continuity and closure of the ileostomy three to six months later. Post-transplant monitoring included endoscopy and biopsy twice weekly for the first 6 weeks, weekly until month three, biweekly until month six, and monthly thereafter, as previously described [14,15].

### 2.3. Post-Transplant Immunosuppression

Immunosuppression consisted of induction with non-depleting antibodies (basiliximab) or depleting antibodies (thymoglobulin) in highly sensitized patients followed by maintenance with tacrolimus, sirolimus and prednisone. Rejection was diagnosed by standard histological criteria in the presence of increased and/or bloody ileostomy output and graded as mild, moderate or severe. Treatment of rejection consisted of steroid pulse therapy, increased tacrolimus levels and/or thymoglobulin depending on the severity of the rejection episode.

### 2.4. Stool Sample Collection

Stool samples were prospectively collected according to the NIH microbiome protocol [5] during routine ambulatory visits and prepared for sequencing and for metabolomic studies as published previously [7,8]. DNA was extracted from stool samples at Georgetown University and all DNA sequencing and bioinformatics conducted at the J. Craig Venter Institute (JCVI) (Rockville, MD, USA).

### 2.5. Microbiome Analysis

16S rRNA gene sequencing of 43 samples was performed with Illumina MiSeq instrument (Illumina, San Diego, CA, USA). For the initial processing of the 16S rRNA datasets, we used the JCVI-developed open-source pipeline YAP (https://github.com/andreyto/YAP, accessed on 17 July 2022) that automates mothur 14 standard operating procedure (www.mothur.org/wiki/MiSeq_SOP, accessed on 17 July 2022). Sequences were aligned against the SILVA rRNA database and assigned taxonomy using the Ribosomal Database Project.15, 16. The statistical analyses were conducted in R, 23 using the open-source package MGSAT (https://github.com/andreyto/mgsat, accessed on 17 July 2022), as previously described [16,17,18,19]. Microbiome data from the Personal Genome Project obtained from the data repository of the American Gut project (https://github.com/biocore/American-Gut/tree/master/data/PGP, accessed on 17 July 2022) was used to represent healthy intestinal microbiome in the comparative analysis (Control group).

### 2.6. Metabolite Extraction and Analysis

Fecal pellets were homogenized on ice in 600 μL of an extraction solution made up of 10% water, 40% methanol, 20% chloroform, 30% isopropanol (IPA), 0.1% Debrisoquine (1 mg/mL in ddH2O) and 0.5% 4-Nitrobenzoic acid (1 mg/mL in Methanol). The samples were incubated on ice for 15 min, then centrifuged at 15,493× *g* for 20 min at 4 °C. The supernatants were transferred to new tubes containing 600 μL of chilled acetonitrile (ACN) and were kept at −20 °C overnight to precipitate protein. The samples were again centrifuged at 15,493× *g* for 20 min at 4 °C, and the supernatants were transferred to new tubes and vacuum dried. Finally, the dried samples were resuspended in 200 μL of 50:50 methanol/water and transferred to LC vials for UPLC-QToF-MS analysis. A quality control sample (QC) was also prepared by pooling an aliquot of each of the prepared samples and was run every 10 samples.

The samples were analyzed by an Acquity UPLC coupled to a Xevo G2 QToF MS (Waters Corporation, Milford, MA, USA). A volume of 2 μL of each sample was injected onto either an Acquity UPLC BEH C18, 130 Å, 1.7 µm, 2.1 mm × 50 mm column maintained at 40 °C for the metabolomic acquisition or a CSH C18, 130Å, 1.7 µm, 2.1 mm × 100 mm column maintained at 55 °C for the lipidomic acquisition. The LC solvents used were 100% water with 0.1% formic acid (A), 100% ACN with 0.1% formic acid (B), 40:60 water/ACN with 0.1% formic acid and 10 mM ammonium formate (C) and 90:90 IPA/ACN with 0.1% formic acid and 10 mM ammonium formate (D). The metabolomic gradient with a flow rate of 0.4 mL/min was set as follows: initial–95% A, 5% B; 0.5 min–95% A, 5% B; 8.0 min–2% A, 98% B; 9.0 min–2% B, 98% D; 10.5 min–2% B, 98% D; 11.5 min–50% A, 50% B; 12.5 min–95% A, 5% B; 13.0 min–95% A, 5% B. The lipidomic gradient had a flow rate of 0.40 mL/min and was run set as follows: initial–60% C, 40% D; 5.0 min–59% C, 41% D; 10.0 min–29% C, 71% D; 11.6 min–1% C, 99% D; 13.0 min–1% C, 99% D; 13.6 min–60% C, 40% D, 15.5 min–60% C, 40% D.

The column eluent was introduced into the QToF MS operating in either positive or negative by electrospray ionization. Positive mode had a capillary voltage of 3.00 kV and a sampling cone voltage of 30 V. Negative mode had a capillary voltage of 2.00 kV and had a sampling cone voltage of 30 V. The desolvation gas flow was set to 1000 L/h, and the desolvation temperature was set to 500 °C. The cone gas flow was 25 L/h and the source temperature was 120 °C. The data was acquired in the sensitivity MS mode with a scan time of 0.300 s and an interscan time of 0.014 s. Accurate mass was maintained by infusing leucine enkephalin (556.2771 m/z) in 50% aqueous acetonitrile (2.0 ng/mL) at a rate of 20 µL/min via the Lockspray interface every 10 s. Data was acquired in Centroid mode with a 50.0 to 1200.0 m/z mass range for TOF MS scanning. The pooled QC was injected every 10 samples to monitor any shifts in retention time and intensities.

### 2.7. Clinical Outcomes

Clinical outcomes were analyzed in terms of patient and graft survival and incidence, timing, and severity of rejection episodes post-transplant. As an indicator of clinically relevant infection complications post-transplant, we analyzed the incidence of bacteremia post-transplant with the assumption that bacteremia episodes post-transplant, in the absence of central venous catheter-derived infection, resulted from gut translocation, which is considered an alteration of the mucosal barrier secondary to rejection.

## 3. Results

### 3.1. Patient and Graft Survival

Overall, 25 of the 43 patients (58%) included in this study remain alive with excellent graft function and on minimal immunosuppression regimens. The survival rate in group A was 75% (17 of 23 patients) with a median follow-up of 13 years (range 6–20) following transplantation. These patients maintain normal graft function and require only minimal immunosuppression regimens. Among patients who died in this group after a median of 10 years from transplant (range 6–12 years), five out of six maintained normal graft function at the time of death. The cause of death in these five patients was due to complications of prolonged immunosuppression in three patients (aplastic anemia, metastatic rectal cancer, and sepsis) and unrelated to transplant in the remaining two patients (car accident, sudden death at home). The remaining patient died following severe acute rejection nine years post-transplant, secondary to non-adherence to immunosuppression medications.

Similarly, overall survival in Group B was 69% (11 of 16 patients), with a median survival of five years (range 1–7 years) following transplantation. These patients maintain normal graft function on minimal immunosuppression regimens. Five patients in this group died from sepsis (*n* = 3) and lung cancer (*n* = 2).

In contrast, all four patients in Group C, including one patient who received a second transplant for chronic rejection of the first transplant performed eight years earlier, experienced severe acute rejection at a median of 5.5 months (range 4–6 months) and suffered multiple infectious complications during the first year post-transplant. All patients in this group experienced graft loss at a median of 17 months (range 12–22) and ultimately expired after a median interval of 19.5 months (range 12–38) post-transplant.

The probability of survival for each group is reported in Figure 1.

### 3.2. Rejection Episodes and Systemic Infection

The timing and severity of rejection episodes and the incidence of bacteremia post-transplant in the three groups of patients are reported in Table 2.

Rejection episodes in Groups A and B tended to be less severe and to occur early after transplant. In contrast, patients in Group C experienced more severe rejection episodes which tended to occur at a later time interval after the transplant (164 days in Group C as compared to 20 and 21 days in Groups A and B, respectively).

Overall, episodes of bacteremia post-transplant were significantly less frequent in Groups A (9%) and B (56%) compared to Group C (100%). Among the episodes of bacteremia, Enterococcus species were isolated from the blood cultures of 50% of patients in Group C compared to none in Group A and in one patient (6%) in Group B.

### 3.3. Fecal Microbiome

The dissimilarity between the microbiome profiles of the three groups of patients and the control group according to redundant analysis is represented in Figure 2.

According to this analysis, the microbiome of the transplanted intestine overall differs from the microbiome of the general population. Among transplant recipients, those with stable allograft function in the long and short term (Group A and B, respectively) show a microbiome profile more similar to one another and closer to the general population, as compared to the microbiome of patients with unstable allograft (Group C).

When analyzed at a phylum level, the similarity of the microbiome of patients in Groups A and B is confirmed, as shown in Figure 3.

In both Groups A and B, there is a large predominance of Firmicutes and a similar composition of other phyla, including Proteobacteria and Bacteroidetes.

At a lower taxonomic level (genus level), the relative bacterial composition, based on 16S rRNA gene sequencing, confirms the differences between the three Groups and the general population (Figure 4). In addition, this analysis demonstrates that the microbiome of patients with stable graft in Group A manifests a closer similarity to the control group than to Groups B or C.

The relative taxonomic abundance at the genus level in the three groups is represented in Figure 5. A decrease in Bacteroides, Prevotella and Faecalibacterium is observed in Groups A, B and C when compared to the control. In contrast, Group C demonstrates a significantly higher level of Enterococcus as compared to Groups A and B and the control.

### 3.4. Metabolomic Analysis

An untargeted metabolomic and lipidomic approach has been adopted in this study to detect changes between patients with stable and unstable allografts. The LC-MS data were preprocessed using the XCMS R and resulted in 2439 positive mode and 1523 negative mode features for the metabolomics run, and 1494 positive and 1102 negative mode features for the lipidomics run. Principle component analysis was performed and a heatmap was generated using samples from all three groups to visualize group separation (Figure 6).

Binary comparisons performed between patients in Groups A and B revealed a total of only 5 metabolites that were significantly downregulated (FC < 0.5 and FDR < 0.05) in the untargeted data (Figure 7). Downregulation of certain annotated phospholipid classes, such as phosphoserines (PS), phosphatidylinositols (PI) and phosphatidylethanolamines (PE) can be seen in the stable at 5 years group. Additionally, dysregulation of several annotated tripeptides can be seen. Downregulation of these or any of the other downregulated metabolites could be early indicators of long-term graft stability, although clear differences between these two groups parallel to the microbiome profiles could not be determined in the metabolic profiles.

## 4. Discussion

In this study, we report the long-term survival up to 21 years for patients who received intestinal transplantation and maintain normal graft function with minimal doses of immunosuppressants. Some of these patients represent the longest worldwide survivors after intestinal transplantation. We report here our results from the analysis of the fecal microbiome and metabolomic profiles of these patients to identify features supportive of prolonged graft stability and as a step toward the identification of non-invasive markers of graft function.

We first demonstrated that the transplant microbiome is overall different from the general population, supporting the notion that the immunosuppressed state after transplant induces changes in the intestinal flora that are observed even several years after transplant. Similar findings have also been reported in lung transplant recipients in whom the load and composition of bacterial microbiome differed significantly from the general population and patients with chronic lung disease [20]. Interestingly, while the gut microbiome has been the focus of ongoing research in kidney [21,22], liver [23,24,25] and intestinal [26] transplant patients, the host-microbiome relationship in transplant recipients remains poorly understood [27], particularly in regard to the impact of immunosuppression on microbiome profiles [28]. While changes in microbiome profiles as a result of different immunosuppression regimens have been reported, suggesting that immunosuppression has an impact on the human microbial population [29], the entity and stability of these changes and their impact on graft function are still incompletely understood and deserve further studies.

We then found that among patients with stable graft function, the microbiome and metabolomic profiles change over time. Specifically, the patterns observed in patients with long-term survival and graft function (5 years or longer) were closer to the general population than the patterns observed early post-transplant (1 year). The causes of this difference remain speculative but are likely related to several factors, including immunosuppression regimens (heavier in the early post-transplant phase compared to long term), use of antimicrobials as routine prophylaxis in the early phase, diet, and others.

Next, we documented a loss of Bacteroides, Prevotella and Faecalibacterium amongst all transplanted intestine. While Group A was most similar to the control group, we also demonstrated a relative increase in Enterococcous and Fuminococcus within Group C when compared to Groups A and B and the control group. Interestingly, a loss of diversity and domination by individual taxa have been reported in association with pathologic states, including inflammatory bowel disease [30], and recently described to be predictive of mortality following allogeneic hematopoietic cell transplantation [31]. The changes that we observed in the microbial composition of patients with rejection are consistent with similar findings reported by Oh et al. [26]. Although their study was performed on ileostomy fluid samples rather than stool samples, they reported a significant decrease of the phyla Firmicutes and Lactobacillales and an increase in Proteobacteria during episodes of rejection.

Our patients in Group C, unlike the other two groups, experienced early post-transplant graft loss despite aggressive treatment. The fecal microbiome in this group showed a dramatic difference from the other two groups and an even larger difference as compared to the general population in terms of species representation. Importantly, Enterococcus species were significantly over-represented in this group. Despite the small group size compared to Groups A and B, the proportional predominance of Enterococcus species compared to other genera in this group was remarkable. This is an important finding, as it correlates with the clinical experience of high morbidity and mortality observed in patients suffering from Enterococcus sepsis post-transplant. A similar finding was documented in patients undergoing stem cell transplantation in whom Enterococcus-dominant dysbiosis was detected as early as 1-month post-transplant and correlated with poor survival [32]. Other prior studies highlighted the morbidity associated with intestinal domination by Vancomycin-resistant Enterococcus (VRE) as a precursor for VRE bacteremia in susceptible patients [33].

Although a cause-effect relationship between enterococcus predominance in the stool and graft failure cannot be unequivocally established given the small sample size, it is intriguing to consider that an abnormal fecal microbial profile (whether the cause or the result of damage to the intestinal mucosa) increases the risk of bacterial translocation and bacteremia. These findings have important clinical correlations and potential implications in terms of patient management. If confirmed by larger studies, it is plausible that future strategies to control the degree of Enterococcus and VRE colonization in the gut could have an important positive impact on patient outcome. Finally, in our metabolomic analysis, although untargeted, we identified different patterns of upregulated metabolites amongst the three groups of patients, mimicking the differentiation between the three groups found with the microbiome analysis. This supports the hypothesis that a different state of the intestinal ecosystem exists, based on features detected in a stool sample, between patients with long-term stable graft function, short-term stable graft function and short-term graft dysfunction and loss. A targeted analysis of upregulated pathways remains to be performed to identify molecules or mediators as potential targets for therapy and warrants further in-depth investigations.

This study has several limitations. The first is related to a single sample per patient at one time point post-transplant. Given the dynamic condition of the gut microbiome over time, it would be important to longitudinally analyze multiple samples from the same patient at different time points to better characterize the evolution of the microbiome over time. The second limitation stems from the restricted focus of our analysis on bacterial species, excluding fungal or viral pathogens, which are likely to play a role in the intestinal ecosystem. Additionally, this study is underpowered due to the small sample size of Group C and the lack of correlation between microbiome profiles and the level of immunosuppression at the time of sample collection.

Despite these limitations, we believe that this study offers a relevant contribution to understanding of host-microbiome interactions by reporting on a unique cohort of patients with the longest worldwide survival after intestinal transplant. We believe that our findings open avenues for further studies on a larger patient population and with an integrated analysis of microbial and metabolomic profiles to further determine the role of individual microbial taxa and of specific metabolomic features as non-invasive biomarkers of intestinal allograft function.

## 5. Conclusions

The fecal microbiome of intestinal transplant patients with long-term graft function closely resembles that of the general population, unlike the profiles of patients with early graft non-acceptance in which marked Enterococcus predominance was observed.

## Figures and Tables

**Figure 1 biomedicines-10-02079-f001:**
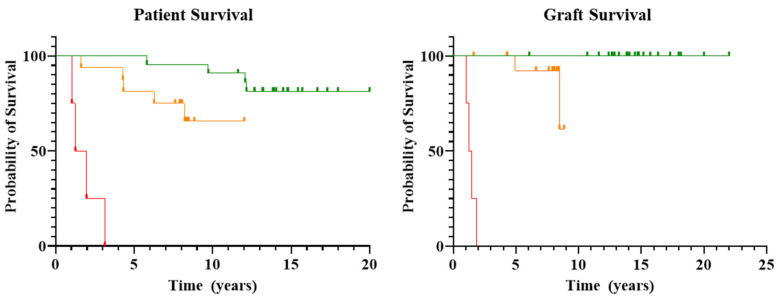
Patient and graft survival. Kaplan-Meier survival analysis was performed for both patient and graft survival of patients in Groups A, B and C. Group A is visually depicted in green, Group B in orange, and Group C in red. Clear differences in the probability of both patient and graft survival can be seen, with Group A having the highest probability of survival, while Group C has the lowest probability of patient and graft survival.

**Figure 2 biomedicines-10-02079-f002:**
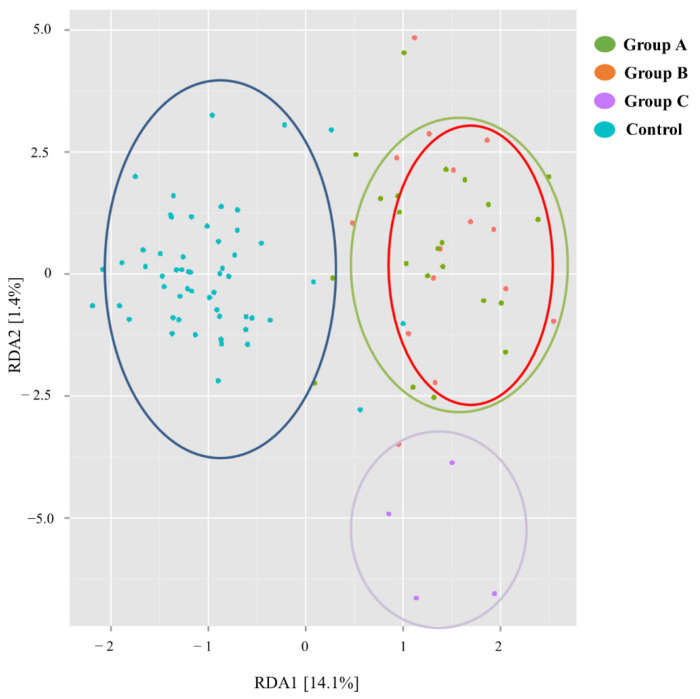
Redundancy Analysis plot comparing the microbiome of the Transplanted Intestine from Groups A, B and C, as compared with the control group. The transplant microbiome overall differs from the general population. Among the three transplant groups, Groups A and B are more similar and closer to the general population than Group C.

**Figure 3 biomedicines-10-02079-f003:**
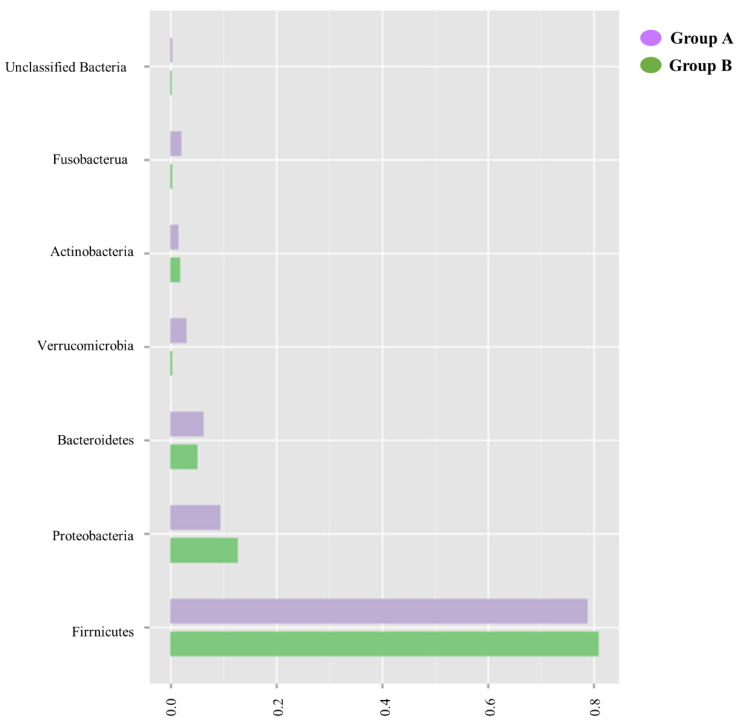
Fecal microbiome analysis comparison of Groups A and B at the phylum level. Overall, the phyla of these groups are similar to each other in composition, with the largest composition being of Firmicutes and the smallest of unclassified bacteria.

**Figure 4 biomedicines-10-02079-f004:**
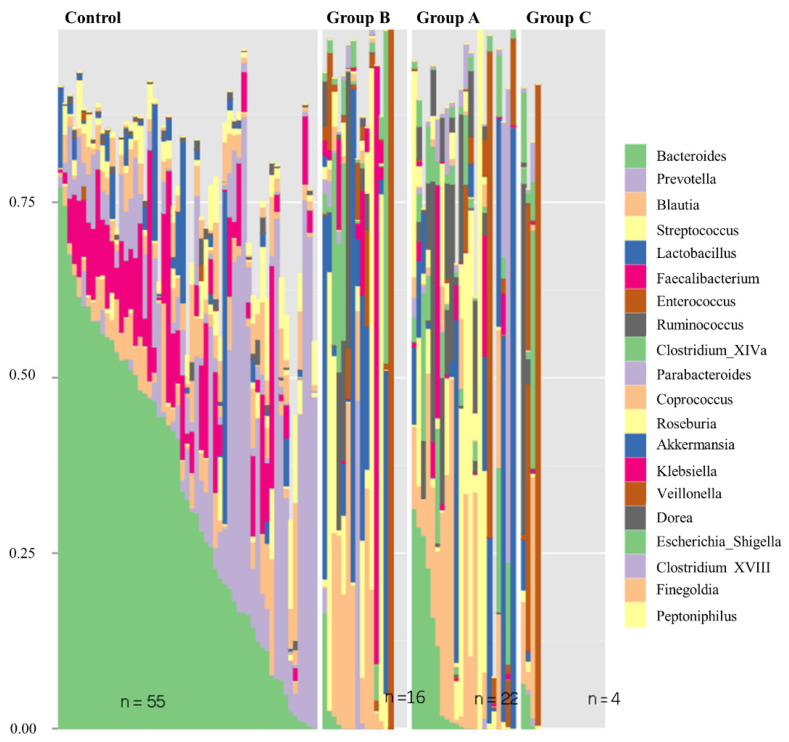
Relative bacterial composition based on 16S rRNA gene sequence classification at the Genus level of taxonomy by individual samples in the control group and the three transplant graft status categories. Increased similarity to the control group is seen amongst Group A than is appreciated in Group B or C.

**Figure 5 biomedicines-10-02079-f005:**
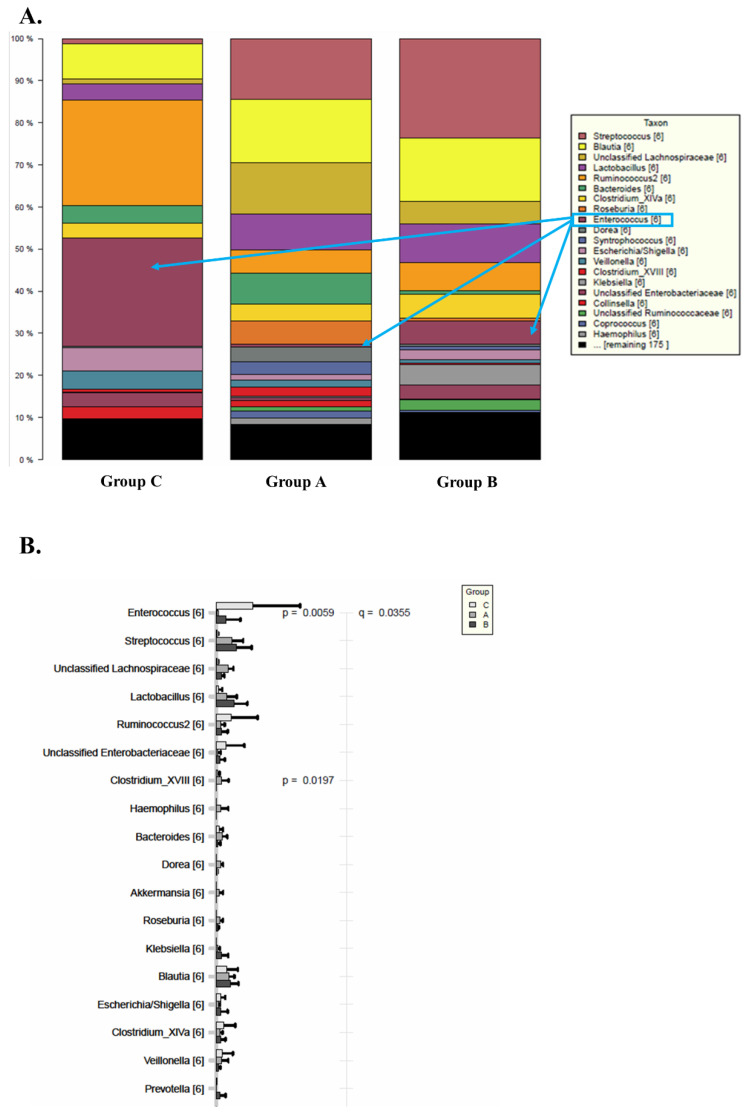
Relative bacterial composition and distribution of genus between Groups A, B and C of the transplanted intestine when compared to the transplanted intestine. (**A**) Histogram demonstrating taxonomic differences. (**B**) Histogram with relevant statistic for taxonomic differences between the groups. Overall, Enterococcus is a more highly expressed in Group C than Groups A and B or the control. Groups A, B and C demonstrate fewer Bacteroides, Prevotella and Faecalibacterium when compared to the control.

**Figure 6 biomedicines-10-02079-f006:**
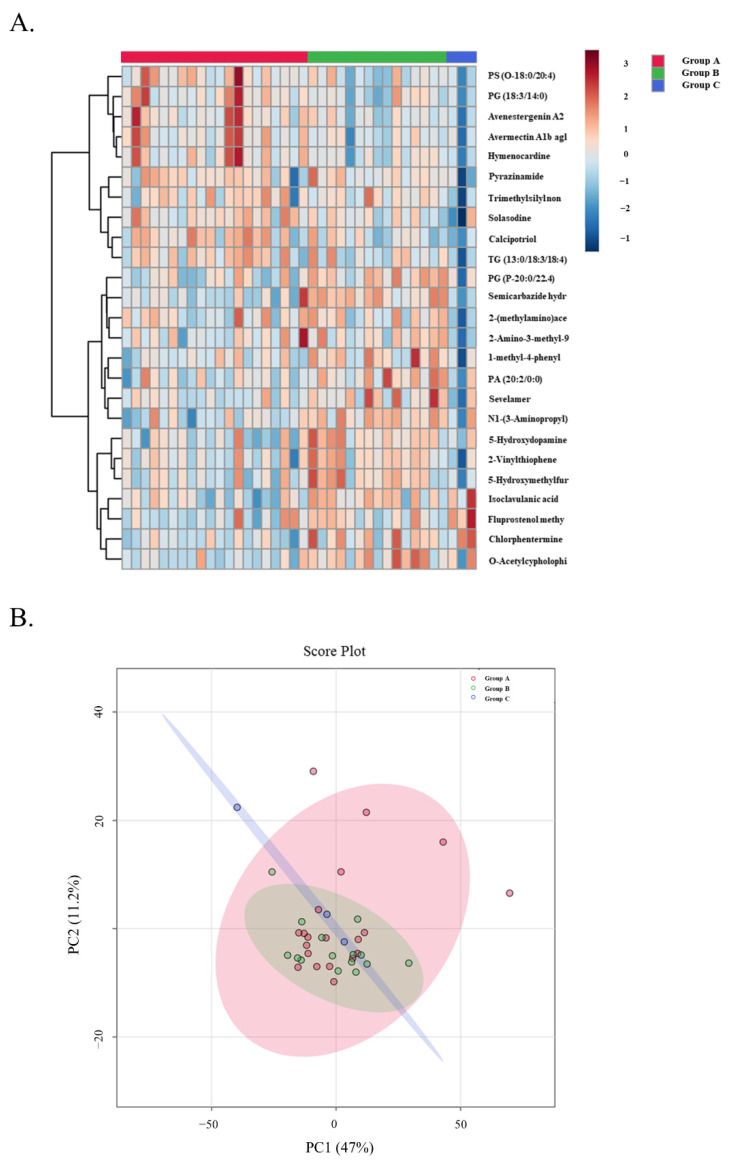
Top 25 metabolites in the three groups. (**A**) LP Heatmap demonstrating the top 25 features in the three groups. (**B**) LP PCA plot demonstrating the top 25 features in the three groups.

**Figure 7 biomedicines-10-02079-f007:**
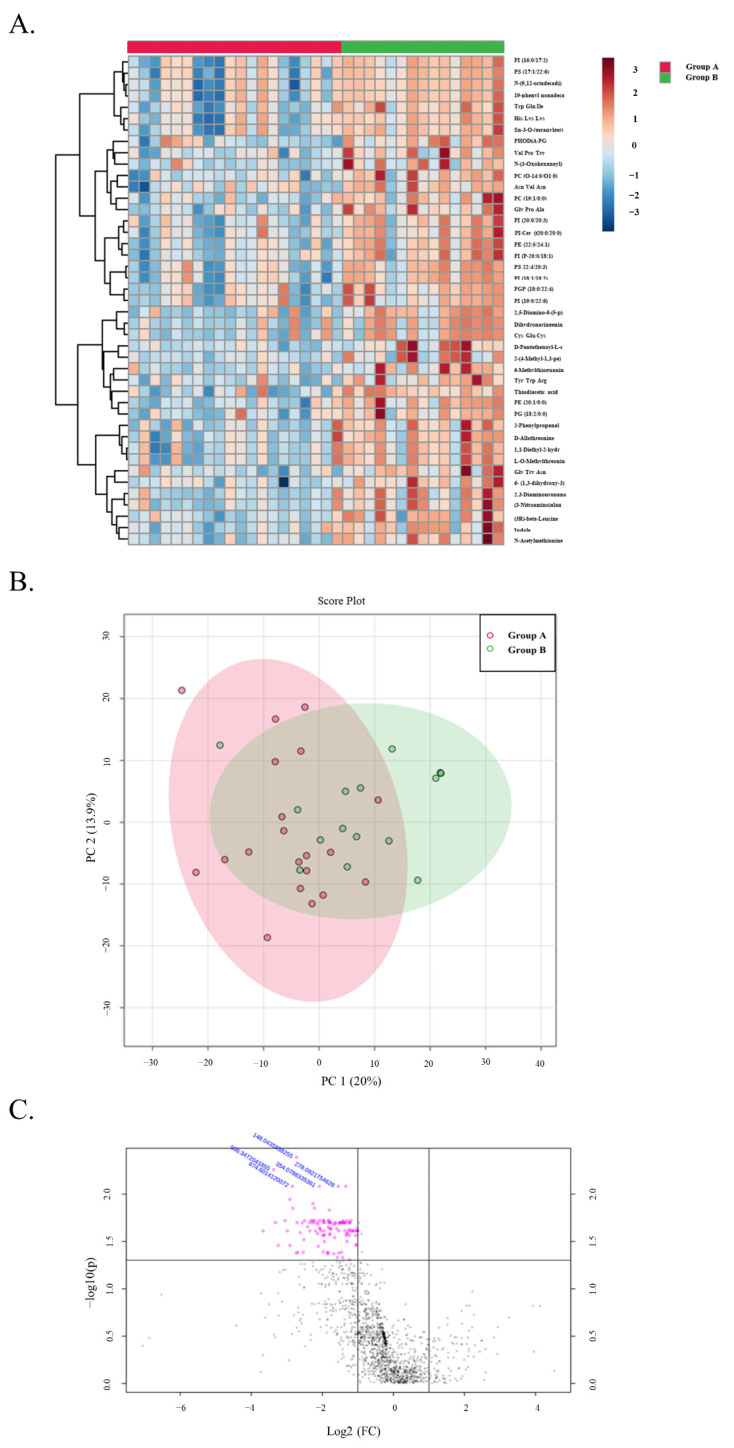
LP Heatmap LP PCA plot: The Top 43 metabolites in group A and B. (**A**) A heatmap visualization showing the relative intensities of metabolomic features in stool samples from patients in Group A versus Group B. Each column represents a patient’s sample, and each row represents a unique feature with a characteristic annotated mass to charge and retention time value. (**B**) A 2D PCA plot showing the overlap between Group A and Group B. (**C**) A volcano plot showing several metabolites found to be significantly (FDR < 0.05) downregulated (Log2FC < 1). The metabolites showing a negative trend could be predictors of long-term graft stability.

**Table 1 biomedicines-10-02079-t001:** Patient Characteristics.

	*Group A* *n* = 23	*Group B* *n* = 16	*Group C* *n* = 4	*p*-Value
**Age at Transplantation (years)**				*p = 0.611*
**Adults**				
Median (range)	42 (19–66)	47 (21–62)	37 (24–53)	
**Pediatric**				
Median (range)	2 (1–15)	2 (1–3)	-	
**Patient Sex [n (%)]**				
Male	15 (65%)	8 (50%)	1 (25%)	*p = 0.275*
Female	8 (35%)	8 (50%)	3 (75%)	*p = 0.691*
**Cause of Intestinal Failure [n (%)]**				*p = 0.286*
**Short Gut Syndrome**	20 (87%)	11 (69%)	3 (75%)	
Gastroschisis	1 (5%)	-	-	
Jejunal Atresia	2 (10%)	-	-	
Mesenteric Thrombosis	3 (14%)	3 (27%)	1 (33.3%)	
Necrotizing Enterocolitis	2 (10%)	-	-	
Radiation Enteritis	-	-	1 (33.3%)	
Surgical Complication	5 (22%)	5 (45%)	-	
Trauma	2 (10%)	1 (9%)	-	
Tumor	1 (5%)	1 (9%)	-	
Volvulus	4 (19%)	1 (9%)	1 (33.3%)	
**Dysmotility**	3 (13%)	5 (31%)	1 (25%)	
Hirsprung’s Disease	-	1 (20%)	-	
Pseudo-obstruction	3 (100%)	4 (80%)	1 (100%)	
**Graft Type [n (%)]**				
Small Intestine	12 (52%)	3 (20%)	1 (25%)	*p = 0.074*
Small Intestine + Colon	3 (13%)	7 (44%)	3 (75%) *	*p = 0.015*
Multivisceral Transplant **	8 (35%)	4 (25%)	-	*p = 0.340*
Modified Multvisceral Transplant ***	-	2 (13%)	-	*p = 0.170*

Age at transplantation was evaluated using one-way ANOVA. All other analysis was performed using the Fisher’s exact test. * One patient in group C received a simultaneous kidney transplant. ** Multivisceral: liver-stomach-duodenum-pancreas-small intestine +/− large intestine. *** Modified Multivisceral: stomach-duodenum-pancreas-small intestine +/− large intestine.

**Table 2 biomedicines-10-02079-t002:** Rejection and Infection.

	*Group A [n (%)]* *n* = 23	*Group B [n (%)]* *n* = 16	*Group C [n (%)]* *n* = 4	*p*-Value
**Acute Rejection**	7 (30%)	4 (25%)	4 (100%)	*p = 0.015*
Mild	3 (43%)	2 (50%)	0	
Moderate	3 (43%)	2 (50%)	3 (75%)	
Severe	1 (14%)	0	1 (25%)	
**Median time from transplant to rejection (days)**	20	21	164	*p = 0.122*
**Bacteremia Episodes**				
*Enterococcus faecium*	0	0	1 (25%)	*p = 0.007*
Vancomycin Resistant Enterococcus	0	1 (6%)	1 (25%)	*p = 0.099*
Other	2 (9%)	8 (50%)	3 (75%)	*p = 0.003*

Acute rejection and bacteremia episodes were analyzed using Fisher’s exact test. Median time from transplant to rejection was analyzed using a one-way ANOVA.

## Data Availability

Not applicable.

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
