# Peer review of "The Microbiome and Metabolomic Profile of the Transplanted Intestine with Long-Term Function"

_biomedicines, 2022, doi:10.3390/biomedicines10092079_

Round 1

Reviewer 1 Report

This is an interesting paper about a very hot topic in gastrointestinal disease and organ transplantation.

I have no major comments.

Minor comments:

Figure 2. is this representing beta-diversity ? should this term be used to homogenize with reporting from other recent studies on GI microbioma ?

Any data about alfa diversity ?

line 196: 25 of the 43.... instead of 43 of the 25 patients (58%)

Author Response

Reply to Reviewer 1.

We appreciate the comments from Reviewer 1. Our reply is as follows:

Figure 2 reports beta-diversity, as a measure of the dissimilarity between the three groups of patients.

In terms of alpha-diversity, this can be appreciated from Figure 4, showing the dissimilarity within groups.

Line 181: 25 of the 43 patients: corrected.

Reviewer 2 Report

major I think this is an excellent  paper and it is a well written and well designed experiment protocol A modern molecular microbiomics approach was used and their conclusions form their data are sound The future directions outlined in discussion are also germane

Minor  section 3.1 the number 43 and 25 are reversed

The university of pittsburgh did an early evaluation of intestinal transplant flora with more primitive quantitiative culture techniques perhaps for historical continuity this could be mentioned in the intro or discussion otherwise i enthusiastically think this paper should be published 

Abu-Elmagd K, Todo S, Tzakis A, Furukawa H, Bonet H, Mohamed H, et al. Intestinal transplantation and bacterial overgrowth in humans. Transplant Proc. 

Author Response

Reply to reviewer 2:

WE appreciate the comments from Reviewer 2. Our reply is as follows:

1-section 3.1 : 25 of 43 patients... has been corrected.

2-the suggested historical paper using culture-based technique has been added to the references